# *Furcanthicus* gen. nov., a New Genus of Oriental Anthicini (Coleoptera, Anthicidae), with Description of Three New Species [note 1]

**DOI:** 10.3390/insects14020102

**Published:** 2023-01-18

**Authors:** Yuchen Zhao, Xinpu Wang, Seunghyun Lee, Ming Bai

**Affiliations:** 1School of Agriculture, Ningxia University, Yinchuan 750021, China; 2Key Laboratory of Zoological Systematics and Evolution, Institute of Zoology, Chinese Academy of Sciences, Beijing 100101, China; 3Research Institute for Agricultural and Life Sciences, Seoul National University, Seoul 151-921, Republic of Korea

**Keywords:** Anthicinae, *Furcanthicus*, new genus, geographic distribution, taxonomy

## Abstract

**Simple Summary:**

A new genus of ant-like flower beetle, *Furcanthicus*, is described along with three new species. Due to a unique combination of morphological characters, we are proposing this new genus separate from *Anthicus*. Three little-known species are redescribed. Habitat photos of some species of the new genus are provided, which is helpful for understanding the lifestyle of these species. An identification key to all species of the new genus is provided. The important morphological characters, distribution, and diagnosis of the new genus are briefly discussed.

**Abstract:**

*Furcanthicus* gen. nov. (Anthicinae: Anthicini) and three new species from the Oriental region are described: *Furcanthicus acutibialis* sp. nov. (China: Tibet), *F. telnovi* sp. nov. (China: Yunnan), and *F. validus* sp. nov. (China: Sichuan). Some critical morphological characters of this genus are discussed. Eight new combinations are established for the following taxa: *Furcanthicus punctiger* (Krekich-Strassoldo, 1931) comb. nov., *F. rubens* (Krekich-Strassoldo, 1931) comb. nov., *F. maderi* (Heberdey, 1938) comb. nov., *F. monstrator* (Telnov, 2005) comb. nov., *F. vicarius* (Telnov, 2005) comb. nov., *F. lepcha* (Telnov, 2018) comb. nov., *F. vicinor* (Telnov, 2018) comb. nov. (all from *Anthicus* Paykull, 1798), and *Nitorus lii* (Uhmann, 1997) comb. nov. (from *Pseudoleptaleus* Pic, 1900). Two informal species-groups are established: *F. maderi* and *F. rubens* species-group. The little-known species *F. maderi, F. rubens*, and *F. punctiger* are redescribed, diagnosed, and illustrated. The distribution map and key to species of this new genus are provided.

## 1. Introduction

Members of Anthicidae Latreille, 1819, are commonly known as “ant-like flower beetles”. The family is a rather diverse group of the superfamily Tenebrionoidea Latreille, 1802, with eight subfamilies [1] and over 3500 extant species [2]. The type of genus of Anthicidae, *Anthicus* Paykull, 1798, comprises various species with high morphological diversity, and is the largest genus of Anthicidae [3]. To improve organization, taxonomists have established species-groups and subgenera for *Anthicus* (for instance, [4,5,6,7,8,9,10]). Subsequently, some of those classifications were gradually promoted into separate genera (for instance [1,11,12,13,14,15]). In addition, in recent years, taxa in Anthicini Latreille, 1819, have been established or revised [3,16,17,18,19,20,21,22,23,24,25], which has contributed to an improved understanding of Anthicini, including the relationships between those genera and *Anthicus*.

When dealing with *Anthicus* collected from China, we found some species with male genitalia different from those in another group from the Old World and more similar to those in the New World genus *Ischyropalpus* LaFerté-Sénectère, 1849. Meanwhile, they seem like *Sapintus* Casey, 1895, at pronotum with basal laterally foveae above procoxae and mesepimera with foveae covered by dense setose. Among them, two new species showed a similar appearance as *Anthicus maderi* Heberdey, 1938, while another new species different from *A. maderi* in the pronotum shape and more similar to *A. rubens* Krekich-Strassoldo, 1931.

The pronotum was a key structure in distinguishing the genera of anthicids by an earlier author, but with the discovery of more species this feature is becoming less distinct and reliable in diagnostics [14]. The morphological characters of mesothorax and male genitalia show more important evolutionary value for genera, families, and subfamilies [16,18,26].

*Anthicus maderi* has been considered an untypical species for the genus *Anthicus*. Heberdey (1938) mentioned in the original description of this species, that it should belong to *Anthicus* group VIII of Marseul (at present, most members of this group belong to *Cordicomus* Pic, 1894) and is similar to *Leptaleus* LaFerté-Sénectere, 1849 (*Pseudoleptaleus* Pic, 1900) [27]. Telnov has described a few new taxa similar to *A. maderi* [21,28,29] and mentioned *A. monstrator* Telnov, 2005, is easily distinguished from other species in the Oriental Region but *A. vicarius* Telnov, 2005 by specific characteristics [28]. Another species, *A. punctiger* Krekich-Strassoldo, 1931, was described as *Anthicus* group IX of Marseul (at present, most of this group members belong to *Stricticomus* Pic, 1894) [30]. After the study of the type specimens of *A. maderi*, *A. punctiger*, *A. rubens* and similar species, we found that they showed a certain stability in mesothoracic and male genital structures.

The purpose of this paper is to describe a new genus for the above species, and the diagnosis of the new genus distinct from other relatives in Anthicini genera is also discussed in the preferred habitats of members of the new genus in China and their distribution.

## 2. Material and Methods

Specimens were collected by sweeping or beating grass and preserved in absolute ethanol. Male genitalia were examined after being cleared in hot 10% KOH solution. Tegmen were examined after added cover slide. Specimens were examined with a Leica M205A stereomicroscope (Leica Microsystems, Wetzlar, Hessian, Germany) and measured using Leica Application Suite v4.12.0 software (Leica Microsystems, Wetzlar, Hessian, Germany). Specimen photographs were taken using a Leica DMC 4500 digital camera (Leica Microsystems, Wetzlar, Hessian, Germany) mounted on the stereomicroscope. Images of the same specimen at different focal planes were combined using Leica Application Suite v4.12.0 and edited with Adobe Photoshop 2022 v23.4.1 (Adobe Systems, San Jose, CA, USA) software. Maps were based on “World Imagery” map of Esri company (https://goto.arcgisonline.com/maps/World_Imagery, accessed on 1 November 2022) and edited using ArcGIS Pro v2.8 (Environmental Systems Research Institute, Redlands, CA, USA). The following abbreviations are used in the text: Elev.—elevation; Ex.—examined. Data from locality labels are cited verbatim for the type specimens only, and comments are placed in square brackets ([]). Separate labels are indicated by double slashes (//). Species distribution information is based on the literature [27,28,29,30,31,32] and specimen labels.

The specimens treated in this study are deposited in the following collections:·IZCAS—Institute of Zoology, Chinese Academy of Sciences, Beijing, China·NHMW—Naturhistorisches Museum Wien, Vienna, Austria·SANXU—the Insect Collection, School of Agriculture, Ningxia University, Ningxia, China

## 3. Results

### 3.1. Furcanthicus Zhao et Wang gen. nov.

urn:lsid:zoobank.org:act:CDE01559-5530-4D00-B2A0-DDE08FCE3EE6

Type species: *Anthicus maderi* Heberdey, 1938, by present designation.

**Description.** Small, total body length usually less than 3.00 mm; surface glossy (at least on elytra). Head ovoid to subtriangular; frontoclypeal suture distinct, nearly straight; compound eyes spherical and entire; antennae filiform, with 11 antennomeres, antennal insertions exposed and clearly visible in dorsal view; mandible apex bidentate, maxillary palp quadrimerous, terminal palpomere securiform. Neck narrow, less than 1/3 of head width across eyes. Pronotum variable in shape; anterior rim and antebasal sulcus of pronotum present, collar of equal width dorsally and ventrally and antebasal sulcus originating at lateral foveae. Mesoventrite slightly expanded, subtriangular, nearly campaniform, its anterior angle slightly rounded, lateral margins either moderately arcuate to mesepisterna in apical third; posterior angle of mesoventrite narrow and round; mesepisterna slightly separated by mesoventrite, and anterior angle of mesoventrite not crossing anterior edge of mesepisterna; posterior angle of mesoventrite and posterior margin of mesepisternum with fringed setation; mesepimera excavated, with setose fringe on margin of cavity. Elytra elongate, undersetae absent; elytra in males lacking apical notches; postbasal impression indicated or not. Abdominal ventrite III without foveae immediately behind metacoxae, intercoxal process triangular; ventrite IX (spiculum) Y-shaped thin and extremely less sclerotized. Male aedeagus entirely open ventrally, with parameres incomplete fused to tegmen, clearly divided into parameral plate and phallobase, phallobase basal margin rounded, and penis slides freely in tegmen.

**Etymology.** Composed from the Latin “*furca*” (fork) and *Anthicus*, referring to the shape of tegmen and relationship with *Anthicus* Paykull, 1798. Masculine gender.

**Diagnosis.** A combination of the following characters of *Furcanthicus* gen. nov., these characters are also found in other genera of Anthicini but never in this combination: (1) Pronotum with basal transverse sulcus continued laterally to foveae above procoxae; (2) mesepisterna with foveae conspicuous, with setose fringe on margin of cavity; (3) abdominal ventrite III without foveae immediately behind metacoxae; (4) ventrite IX (spiculum) thin and extremely less sclerotized (Figures 1–5I); and (5) male aedeagus entirely open ventrally, with the parameres incomplete fused to tegmen, phallobase basal margin rounded (Figures 1–6G,H).

**Biology.** Adults of Chinese species mostly collected by sweeping or beating on monocotyledonous bunchgrass or bamboo leaf, rarely collected on dicotyledonous vegetation or living (without dead branches) shrubs, and some specimens were attracted to light. They prefer to hide in the center of bunchgrass and walk on grass or stones/ground under the grass. So far, it has not been found on flowers and does not appear to be attracted to cantharidin. 

**Distribution.** Up to now, most members of this genus have been recorded in the Himalayas and Oriental Region, with the northernmost record in Pakistan (Gojal) (Figure 7). One species, *Anthicus lii* (Uhmann, 1997), collected from central China, was recorded as being similar to *Anthicus maderi* based on the original description and drawings [21] and is here transferred to *Nitorus lii* (Uhmann, 1997) comb. nov. according to the type of material.

**Remarks.***Furcanthicus* gen. nov. belongs to the subfamily Anthicinae Latreille, 1819, tribe Anthicini Latreille, 1819, due to the combination of the following characters: (1) Head with frontoclypeal suture present, clypeus transverse, anterior margins of eyes rounded to briefly flattened, neck narrow; (2) pronotum with anterior collar of equal width dorsally and ventrally, pronotal horn not present, antebasal sulcus of pronotum present, procoxal cavities open externally but closed internally; (3) mesoventrite subtriangular with distinct suture marking lines of fusion with mesepisterna; (4) metacoxae moderately separated by intercoxal process of the first visible abdominal sternite margins, with marginal bead exclusively complete; (5) elytra with sutural stria present on at least apical third; (6) tibial spurs spinulose, tarsal claws simple; (7) male genitalia with free and distinct penis and tegmen open ventrally, clearly divided into parameral plate and phallobase.

### 3.2. Furcanthicus maderi Species-Group

**Diagnosis.** Surface glossy, pubescence yellowish, sparse, and appressed. Pronotum generally not wider than head, broadly rounded anteriorly and laterally, moderately narrowed posteriad, antebasal sulcus present (Figures 1–4C), broad laterally. Protibiae and metatibiae of male usually with processes or curved (Figure 3J).

#### 3.2.1. *Furcanthicus maderi* (Heberdey, 1938) comb. nov.

(Figures 1 and 9C,E)

Original placement: *Anthicus maderi* Heberdey, 1938: 163.

**Type material.** Lectotype ♂, Vallis flumin. Soling-ho, Yun. [printed]//coll. Heberdey [printed]//Maderi nov. [handwritten] det. Dr. R. F. Heberdey [printed]//TYPE [printed, label red]//LECTOTYPUS [printed, label red, designated by Telnov] (NHMW).

**Additional specimens.** 1♂, E-Lu Park (101.5338° E, 25.0302° N), Chuxiong City, Yunnan, China, elev. 1856 m, 12.VI.2017, Yuchen Zhao (SANXU); 2♂♂, 1♀, Dali City (100.2583° E, 25.5937° N), Yunnan, China, elev. 2002 m, 15.VI.2017, Yuchen Zhao (SANXU); 2♂♂, 2♀♀, Kunming City, Yunnan, China, 16.V.2015, Zhangxun Wang (SANXU); 1♂, Longxing Village (98.6572° E, 24.6603° N), Baoshan City, Yunnan, China, elev. 1340 m, 27.V.2021, Yuchen Zhao (SANXU); 87 ex., Luozizuo Village (100.2173° E, 25.1142° N), Weishan County, Dali City, Yunnan, China, elev. 2364 m, 07.VII.2021, Yuchen Zhao and Xinpu Wang (SANXU); 1♂, Wuding County (102.4208° E, 25.5408° N), Chuxiong City, Yunnan, China, elev. 1717 m, 19.V.2022, Yuchen Zhao (SANXU); 1♀, Madeng Village (99.9096° E, 26.5895° N), Jianchuan County, Dali City, Yunnan, China, elev. 2395 m, 01.VI.2022, Yuchen Zhao (SANXU); 1♂, Dadiji Country (101.1670° E, 24.7124° N), Chuxiong City, Yunnan, China, elev. 2073 m, 08.VI.2022, Yuchen Zhao (SANXU).


**Redescription**


**Body.** Total length 2.20–2.71 mm.

**Color.** Glossy, head and pronotum yellowish-brown; elytra black, with wide yellow band on postbasal impression; yellow band reaching lateral margin and not interrupted on suture; antennomeres I-VII yellow, apical four antennomeres gradually darkening to black; mouthparts and legs yellowish-brown, pronotal base, tarsi, base of femora, and apices of tibia paler; ventral forebody yellowish-brown, abdominal ventrites brownish-black (Figure 1A).

**Head.** Smooth, 1.1 times as long as wide, evenly rounded posteriorly; frontoclypeal suture present; tempora longer than a diameter of eye, and converging toward rounded base; temporal angles indistinct; eyes medium-sized, moderately convex, interfacetal setae short; punctures of head sparse and shallow, absent at median of head; setation subdecumbent, light, pointing to the median; tactile setae long, erect (Figure 1B); antennae filiform, extending beyond base of elytra; antennomeres I-VII yellow, I slightly widened, II shorter than III, III-VII similar length; antennomere X longer than wide, XI elongate cylindrical with apex pointed, 1.8 times as long as the penultimate.

**Pronotum.** Longer than wide, 1.3 times as long as wide; with anterior rim and antebasal sulcus distinct; pronotum rounded anteriorly and laterally, slightly constricted posterolaterally then slightly expanding before base in dorsal view; punctures on disc larger and denser than those on lateral sides and on head; pubescence yellow, appressed, directed to disc on lateral sides and posteriorly on disc; tactile setae erect (Figure 1C).

**Scutellar shield.** Triangular, slightly rounded distally.

**Elytra.** Glossy, postbasal impression shallow; 1.7 times as long as wide; elytra with wide yellow band on postbasal impression; elytra base black above yellow band, the black area not interrupted on suture and not reaching to lateral margin; apical 2/3 of elytra black; elytral humeri distinct, broadly rounded; punctures moderately dense, with intervening spaces 2–3 times as large as punctures, punctures gradually smaller and shallower toward apex; pubescence yellow, long, sparse, subdecumbent, directed posteriorly; tactile setae erect, length as setation (Figure 1J). Metathoracic wings fully developed.

**Legs.** Yellowish-brown, with femora slightly thickened distally; tarsi, base of femora, and apices of tibia paler; basal metatarsomere as long as combined length of remaining metatarsomeres.

**Venter.** Glossy, forebody yellowish-brown, abdominal sternites brownish-black; punctures on mesoventrite large and shallow, smooth; setation sparse, subdecumbent, paler, directed posteriorly; mesepimera, posterior angle of mesoventrite and posterior margin of mesepisternum with fringed setation (Figure 1D); last visible ventrite of male rounded distally (Figure 1E); male tergum VII strongly dented distally (Figure 1F).

**Aedeagus.** Tegmen bilobated apically; parameral plate 1.3 times as long as phallobase; lateral margin of penis apex contracted uniformly (Figure 1G,H).

**Sexual dimorphism.** Male protibiae modified, with small, blunt, dent-like protrusion at midlength of inner side, setae dense from here to apex. Female externally similar to male while protibiae simple.

**Ecology.** This species was mostly collected by sweeping or beating grass on paths beside forests or open space from China at elevations of 1300–2500 m (Figure 9C,E).

**Distribution.** China (Yunnan), Thailand (Mae Hong Son).

**Figure 1 insects-14-00102-f001:**
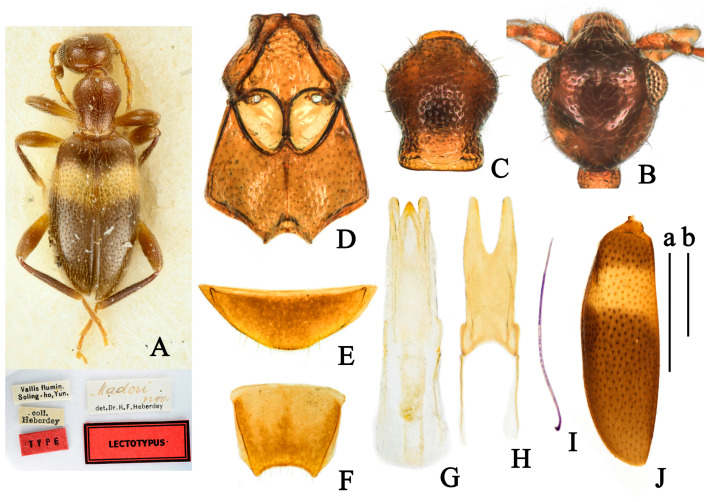
*Furcanthicus maderi* (Heberdey, 1938): (**A**) habitus and labels of lectotype (photography by Dr. Harald Schillhammer); (**B**) head, male, dorsal view; (**C**) pronotum, male, dorsal view; (**D**) meso- and metasternum, male, ventral view; (**E**) ventrite VII, male, ventral view; (**F**) tergum VII, male, dorsal view; (**G**) aedeagus, dorsal view; (**H**) tegmen, dorsal view; (**I**) spiculum, male; (**J**) left elytron, male, dorsal view. Scale bar: (a) 0.5 mm for (**B**–**I**); (b) 0.5 mm for (**J**).

#### 3.2.2. *Furcanthicus punctiger* (Krekich-Strassoldo, 1931) comb. nov.

(Figures 2 and 9D,E)

Original placement: *Anthicus punctiger* Krekich-Strassoldo, 1931: 23.

**Type material.** Type ♂, Nainital, Kumaon, U.P. India. H.G.C. [printed]//coll. Heberdey [printed]//punctiger kr. [handwritten] det. v. Krekich [printed]//TYPE [printed, label red] (NHMW).

**Additional specimens.** 2♂♂, 2♀♀, E-Lu Park (101.5338° E, 25.0302° N), Chuxiong City, Yunnan, China, elev. 1856 m, 12.VI.2017, Yuchen Zhao (SANXU); 3♂♂, 2♀♀, Lucheng Village (101.5352° E, 25.0024° N), Chuxiong City, Yunnan, China, elev.1859 m, 13.VI.2017, Yuchen Zhao (SANXU); 54 ex., Luozizuo Village (100.2173° E, 25.1142° N), Weishan County, Dali City, Yunnan, China, elev. 2364 m, 07.VII.2021, Yuchen Zhao and Xinpu Wang (SANXU); 1♀, Yipinglang Village, Chuxiong City, Yunnan, China, 25.VII.2014, Zhangxun Wang (SANXU); 1♂, Kunming City, Yunnan, China, 16.V.2015, Zhangxun Wang (SANXU); 3♂♂, Qianchang Village (101.3704° E, 25.5372° N), Wuding County, Chuxiong City, Yunnan, China, elev. 2518 m, 23.V.2022, Yuchen Zhao (SANXU); 1♂, 2♀♀, Jianchuan County (99.7553° E, 26.5229° N), Dali City, Yunnan, China, elev. 2448 m, 31.V.2022, Yuchen Zhao (SANXU); 2♂♂, Madeng Village (99.9096° E, 26.5895° N), Jianchuan County, Dali City, Yunnan, China, elev. 2395 m, 01.VI.2022, Yuchen Zhao (SANXU); 1♂, 2♀♀, Pingpo Village (100.0514° E, 25.5578° N), Dali City, Yunnan, China, elev. 1834 m, 06.VI.2022, Yuchen Zhao (SANXU).


**Redescription.**


**Body.** Total length 1.79–2.24 mm.

**Color.** Glossy, yellowish-brown; antennae basal yellow, gradually darkening toward apex; pronotum basal, elytra, tarsi, and tibiae apical yellow; ventral forebody yellowish-brown (Figure 2A). 

**Head.** Round, glossy, length as long as width, evenly rounded posteriorly, with basal margin broadly rounded; frontoclypeal suture straight; tempora as long as a diameter of eye, subparallel, then converging toward rounded base; temporal angles rounded; eyes medium-sized, convex, interfacetal setae short; punctures of head large and shallow, absent at median of basal half; setation decumbent, yellow, point to the median; tactile setae long, erect (Figure 2B); antennae filiform, extending beyond base of elytra; first antennomere slightly widened, II antennomere shorter than III, III-VI with similar length, VII-X gradually widened, penultimate antennomere length longer than width, terminal antennomere elongate cylindrical with apex pointed, 2 times as long as X antennomere.

**Pronotum.** Longer than wide, 1.12 times as long as wide, rounded anteriorly and laterally, slightly constricted posterolaterally in dorsal view, pronotum with anterior rim and antebasal sulcus distinct; punctures of pronotum large, dense, irregular round, nearly confluent; pubescence yellow, appressed, directed posteriorly; tactile setae long, erect (Figure 2C).

**Scutellar shield.** Triangular, slightly rounded distally.

**Elytra.** Glossy, uniform yellowish-brown, parallel sided on base, postbasal impression shallow, 1.9 times as long as wide; humeri distinct, broadly rounded; punctures obviously, with intervening spaces 1–2 times as large as punctures, punctures gradually smaller and shallower toward apex; pubescence pale, long, sparse, subdecumbent, directed posteriorly; tactile setae erect, length as setation (Figure 2J). Metathoracic wings fully developed.

**Legs.** Yellowish-brown; femora slightly enlarged apically but not stick-like; basal metatarsomere slightly shorter combined length of remaining metatarsomeres.

**Venter.** Glossy, yellowish-brown; punctures on mesoventrite large and shallow, smooth; setation sparse, subdecumbent; mesepisternum with setation long and dense; mesepimera, posterior angle of mesoventrite and posterior margin of mesepisternum with fringed setation (Figure 2D), last visible ventrite of male rounded distally (Figure 2E); male tergum VII weakly emarginate distally (Figure 2F).

**Aedeagus.** Tegmen bilobated apically, length of parameral plate hardly longer than phallobase, lateral margin of penis apex drastically contracted and anteriorly pointed (Figure 2G,H).

**Sexual dimorphism.** Male protibiae and metatibia curved inward. Female externally similar to male while protibiae and metatibiae simple.

**Variation.** Body length in the range of 1.68–2.03 mm. Some specimens collected from China are lighter in color and uniformly yellow.

**Ecology.** The habitat of *F. punctiger* from China is similar to that of *F. maderi*. They were mostly collected by sweeping or beating grass on path beside forest or open space at elevations of 500–3100 m (Figure 9D,E). Label record “Low herbage in open oak forest” according to Telnov [31].

**Distribution.** China (Yunnan), India (Sikkim, Uttaranchal), Nepal (Bagmati, Kathmandu).

**Figure 2 insects-14-00102-f002:**
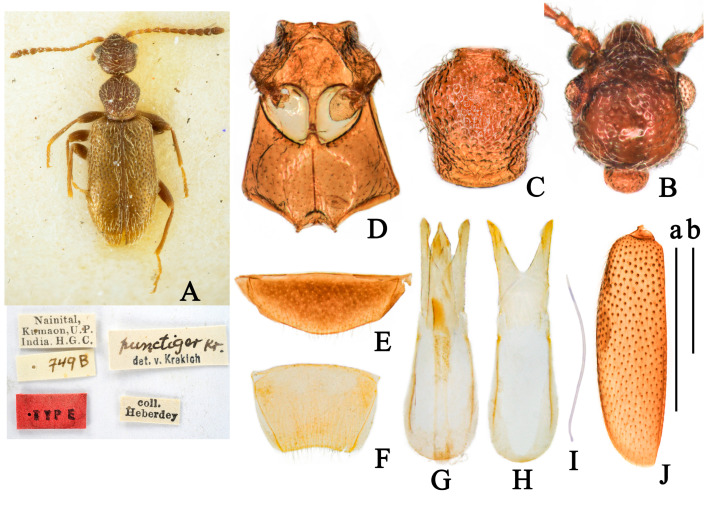
*Furcanthicus punctiger* (Krekich-Strassoldo, 1931): (**A**) habitus and labels of holotype (photography by Dr. Harald Schillhammer); (**B**) head, male, dorsal view; (**C**) pronotum, male, dorsal view; (**D**) meso- and metasternum, male, ventral view; (**E**) ventrite VII, male, ventral view; (**F**) tergum VII, male, dorsal view; (**G**) aedeagus, dorsal view; (**H**) tegmen, dorsal view; (**I**) spiculum, male; (**J**) left elytron, male, dorsal view. Scale bar: (a) 0.5 mm for (**B**–**I**); (b) 0.5 mm for (**J**).

#### 3.2.3. *Furcanthicus acutibialis* Zhao et Wang sp. nov.

(Figures 3 and 9A)

urn:lsid:zoobank.org:act:2C23B6D1-31EB-4A29-B5D7-903AFB2DBC71

**Holotype.** ♂, Zhu Village (95.4319° E, 29.4846° N), Mêdog County, Nyingchi City, Tibet, China, elev. 1654 m, 18.VI.2021, Yuchen Zhao (SANXU);

**Paratypes.** 15♂♂, 13♀♀, same data as holotype (10♂♂, 8♀♀ SANXU; others IZCAS).


**Description.**


**Body. Holotype.** Total length 2.08 mm, maximum width 0.64 mm; head length 0.45 mm, head width across eyes 0.41 mm, pronotal length 0.43 mm, maximum pronotal width 0.37 mm, minimum pronotal width 0.29 mm, elytral length 1.19 mm, combined elytral width 0.64 mm.

**Color.** Body glossy, brownish-black; pronotum paler than elytra; antenna yellowish-brown on base part, gradually darkening toward apex; legs and maxillary palpus brown; basal tarsomere, base of femora and apices of tibiae paler; ventral brownish-black (Figure 3A).

**Head.** Brownish-black, glossy, round, with base margin rounded, slightly flattened in lateral view; tempora longer than a diameter of eye, slightly converging toward rounded base, temporal angles indistinct; eyes medium-sized, moderately convex, interfacetal setae distinct; frontoclypeal suture present; punctures on head large and shallow, with intervening spaces shorter than one diameter of punctures, sparser at median of head; setation of head subdecumbent, brownish, pointing towards different directions; tactile setae inconspicuous (Figure 3B); antennae yellowish-brown, filiform, extending beyond base of elytra; first antennomere brown, slightly widened; II-VI yellowish-brown (some paratypes with antennomeres II-IV or only II yellowish-brown) with similar length, VII-X gradually widened and darkened, antennomere X hardly longer than width, antennomere XI elongate cylindrical, 1.9 times as long as the penultimate.

**Pronotum.** Brownish-black, slightly lighter than head and elytra, glossy, 1.2 times as long as wide, disc flattened in lateral view, with anterior rim and antebasal sulcus distinct, rounded anteriorly and laterally, slightly constricted posterolaterally then slightly extended before base in dorsal view; punctures on pronotum larger and denser than those on head, with intervening spaces 0.2–0.5 times as large as punctures, punctures on side somewhat smaller but also dense; pubescence denser than those on head, brown, subdecumbent, indistinctly directed posteriorly; tactile setae erect, inconspicuously (Figure 3C).

**Scutellum.** Triangular, slightly rounded distally.

**Elytra.** Uniform brownish-black, glossy, postbasal impression absent; humeri distinct, broadly rounded; elytra conjointly rounded apically; punctures of elytra uniform, somewhat dense, with intervening spaces 1–2 times as large as punctures; punctures gradually smaller and shallower toward apex; pubescence yellow, long, sparse, subdecumbent, indistinctly directed posteriorly; tactile setae suberect, length as setation (Figure 3K). Metathoracic wings fully developed.

**Legs.** As brownish-black as pronotum, basal tarsomere, base of femora and apices of tibiae paler; femora slightly enlarged but not stick-like; male protibiae modified, with obvious, blunt, dent-like protrusion at inner side near basal third, setae dense from here to apex (Figure 3J); basal metatarsomere hardly longer than combined length of remaining metatarsomeres.

**Venter.** Glossy, forebody brownish-black, and black everywhere else; punctures of mesoventrite large and shallow, smooth, irregular; setation of mesoventrite sparse, subdecumbent, paler, directed posteriorly; mesepisterna slightly separated by mesoventrite (Figure 3D); last visible ventrite of male inward depression centrally, apical margin bulged medially (Figure 3E); male tergum VII weakly emarginate distally (Figure 3F).

**Aedeagus.** Tegmen bilobated apically; parameral plate 1.68 times as long as phallobase, lateral margin of penis apex contracted uniformly (Figure 3G,H).

**Sexual dimorphism.** Male metatibia not obviously curved inward but more curved than that in females; Female with protibiae and metatibiae simple, and last visible ventrite broadly rounded distally.

**Variation.** Body length in the range of 1.93–2.21 mm. Variable coloration of antenna, II-IV or II-VI antennomeres yellowish-brown, or gradually darkening from 2nd toward terminal antennomere.

**Diagnosis.** This species resembles *Furcanthicus monstrator* (Telnov, 2005) (China: Yunnan and Sichuan) and *Furcanthicus vicarius* (Telnov, 2015) (China: Yunnan and Sichuan), different from *F. monstrator* in with metathoracic wings fully developed and different from *F. monstrator* and *F. vicarius* in the shape of the male protibiae and ventrite VII.

**Etymology.** Named from the Latin word ‘*acutus*’ [sharp] + ‘*tibialis*’ [tibial] because of the modified male protibiae.

**Ecology.** This species was collected by beating grass on the cliff of the hillside path at elevations of 1654 m (Figure 9A).

**Distribution.** Only known from Mêdog County, Tibet, China.

**Figure 3 insects-14-00102-f003:**
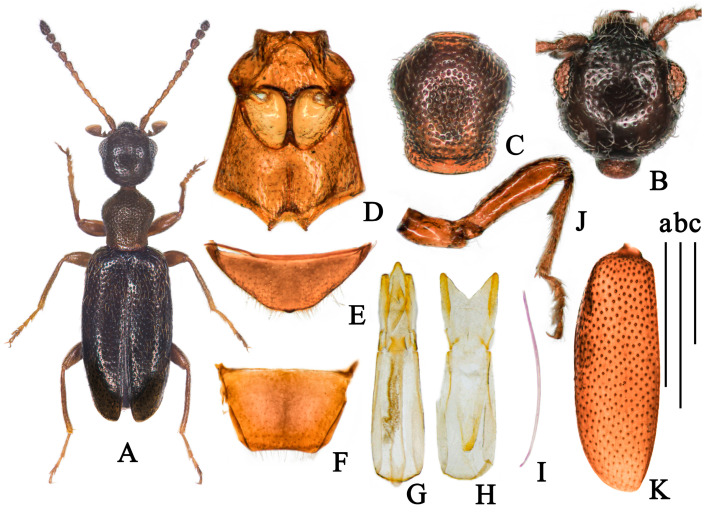
*Furcanthicus acutibialis* sp. nov.: (**A**) habitus of holotype; (**B**) head, male, dorsal view; (**C**) pronotum, male, dorsal view; (**D**) meso- and metasternum, male, ventral view; (**E**) ventrite VII, male, ventral view; (**F**) tergum VII, male, dorsal view; (**G**) aedeagus, dorsal view; (**H**) tegmen, dorsal view; (**I**) spiculum, male; (**J**) left leg, male, inner view; (**K**) left elytron, male, dorsal view. Scale bar: (a) 1 mm for (**A**); (b) 0.5 mm for (**B**–**J**);(c) 0.5 mm for (**K**).

#### 3.2.4. *Furcanthicus telnovi* Zhao et Wang sp. nov.

(Figures 4 and 9C)

urn:lsid:zoobank.org:act:3FD742B5-9542-422B-B90C-88D102FE1D1B

**Holotype.** ♂, Longxing Village (98.6572° E, 24.6603° N), Baoshan City, Yunnan, China, elev. 1340 m, 27.V.2021, Yuchen Zhao (SANXU).

**Paratypes.** 8♂♂, 4♀♀, same data as holotype (SANXU).


**Description.**


**Body. Holotype.** Total length 2.15 mm, maximum width 0.66 mm; head length 0.45 mm, head width across eyes 0.44 mm, pronotal length 0.45 mm, maximum pronotal width 0.39 mm, minimum pronotal width 0.27 mm, elytral length 1.25 mm, combined elytral width 0.66 mm.

**Color.** Body glossy, head and pronotum orange, elytra black, each elytron with wide yellow band near the basal third, antennae yellow in antennomeres I–VI, gradually darkening to black toward apex; mouthparts and legs orange, apices of antennae and bases of tibiae darker (Figure 4A); ventral forebody and first visible ventrite orange, ventrites IV–VII black.

**Head.** Glossy, 1.2 times as long as wide, evenly rounded posteriorly; frontoclypeal suture present; tempora slightly longer than a diameter of eye, and converging toward rounded base, temporal angles indistinct; eyes moderately large, convex, interfacetal setae short; punctures of head sparse and shallow, absent at median of head; setation subdecumbent lighter, point to the median of head base; tactile setae erect (Figure 4B); antennae filiform, extending beyond base of elytra; antennomeres I–VI yellow, I slightly widened, II shorter than III, antennomeres III–V with similar length, VII–XI black, X antennomere length longer than width; XI elongate cylindrical with apex pointed, 1.7 times as long as the penultimate.

**Pronotum.** Pronotum 1.15 times as long as wide, rounded anteriorly, constricted posterolaterally then slightly expanding before base in dorsal view; pronotum with anterior rim and antebasal sulcus distinct, punctures on disc larger and denser than those on lateral sides and on head; pubescence denser than those on head, brownish-black, appressed, directed to disc on lateral sides and posteriorly on disc; tactile setae long and erect (Figure 4C).

**Scutellum.** Triangular, slightly rounded distally.

**Elytra.** Glossy, postbasal impression indistinct, 1.9 times as long as wide, elytra with wide yellow band near the basal third; elytra base black above yellow band, the black area interrupted on suture and not extended to lateral margin of elytra, apical half of elytra black, extended lateral margin and suture; elytral with humeri distinct, broadly rounded; punctures of elytra uniform, sparse, with intervening spaces 1–2 times as large as punctures; punctures gradually smaller and shallower toward the apex; pubescence yellow, long, sparse, subdecumbent, somewhat directed posteriorly; tactile setae erect, thick, length as setation (Figure 4J). Metathoracic wings fully developed.

**Legs.** As orange as pronotum, base of tibiae darker; femora slightly thickened distally not stick-like; basal metatarsomere hardly longer than combined length of remaining metatarsomeres.

**Venter.** Glossy, forebody and ventrite III orange, black everywhere else; mesoventrite lateral margins slightly arcuate, extend in the apical third and near basal margin; punctures on mesoventrite small, indistinct, setation sparse; mesepisterna slightly separated by mesoventrite; mesepimera, posterior angle of mesoventrite and the posterior margin of mesepisternum with fringed setation (Figure 4D).

**Aedeagus.** Tegmen bilobated apically, lateral margin of parameral plate slightly parallel, parameral plate 1.35 times as long as phallobase; lateral margin of penis apex contracted uniformly (Figure 4G,H).

**Sexual dimorphism.** Male protibiae modified, with small, blunt swelling at middle inner side, setae dense from here to apex; the inner side of metatibia emarginate at apical third, setae dense from here to apex; last visible ventrite of male rounded distally (Figure 4E); male tergum VII strongly dented distally (Figure 4F). Female with simple pro- and metatibiae, and last visible ventrite broadly rounded distally.

**Variation.** Body length in range of 1.98–2.24 mm. A specimen of paratypes paler in color than the holotype with all legs yellowish and only the last 4 antennomeres being slightly darker. The black area of elytral apex not extended lateral margin of elytra but still extends to suture.

**Diagnosis.** This species strongly resembles *F. maderi* (Heberdey, 1938) from Thailand and China (Yunnan), *F. lepcha* (Telnov, 2018) from India (Sikkim), and *F. vicinor* (Telnov, 2018) from Nepal. It can be easily distinguished from *F. maderi* according to the coloration of forebody and elytra, distinguished from *F. lepcha* with larger eyes, shorter tempora and without shallow postbasal transverse impression on elytra.

**Etymology.** This species is named after Dr. Dmitry Telnov to demonstrate our sincere respect for his numerous contributions to anthicid taxonomy, especially his contribution to the study of *maderi* species-group.

**Ecology.** This species was collected by sweeping bamboo leaf on roadsides at elevations of 1340 m (Figure 9C).

**Distribution.** Only known from Baoshan City, Yunnan, China.

**Figure 4 insects-14-00102-f004:**
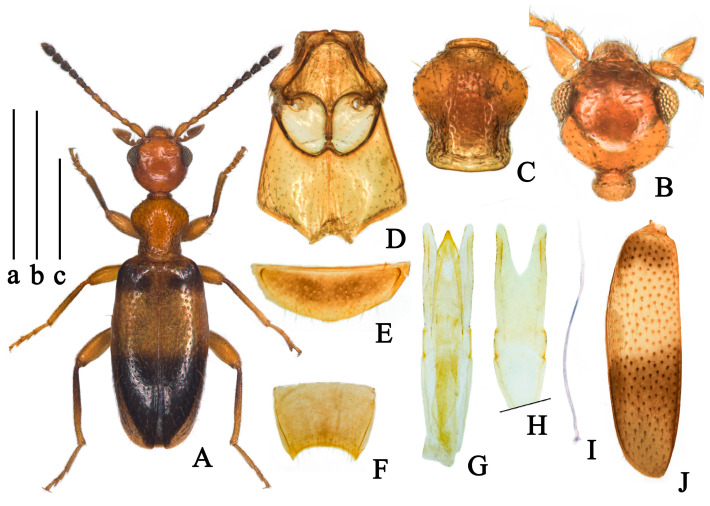
*Furcanthicus telnovi* sp. nov.: (**A**) habitus of holotype; (**B**) head, male, dorsal view; (**C**) pronotum, male, dorsal view; (**D**) meso- and metasternum, male, ventral view; (**E**) ventrite VII, male, ventral view; (**F**) tergum VII, male, dorsal view; (**G**) aedeagus, dorsal view; (**H**) tegmen, dorsal view, base broken; (**I**) spiculum, male; (**J**) left elytron, male, dorsal view. Scale bar: (a) 1 mm for (**A**); (b) 0.5 mm for (**B**–**I**); (c) 0.5 mm for (**J**).

### 3.3. Furcanthicus rubens Species-Group

**Diagnosis.** Small sized, Surface glossy, usually with large punctures. Pubescence yellowish, dense, and appressed. Head square to triangular, tempora subparallel to somewhat diverging, head base straight, with posterior temporal angles distinct (Figure 5B and Figure 6B). Pronotum generally wider than head across eyes, broadly rounded anteriorly, obvious lateral expansion in the apical third, moderately narrowed posteriad, antebasal sulcus present but indistinct (Figure 5C and Figure 6C). Elytra conjointly rounded apically. Protibiae and metatibiae of male usually simple.

#### 3.3.1. *Furcanthicus rubens* (Krekich-Strassoldo, 1931) comb. nov.

(Figures 5 and 9D,E)

Original placement: *Anthicus rubens* Krekich-Strassoldo, 1931: 20

**Type material.** Type ♂, Dhelu, Mandi, Punjab. 4500ft. H.G.C. [printed]//♂ [handwritten]//A. rubens kr. [handwritten] det. v. Krekich [printed]//TYPE [printed, label red] (NHMW).

**Additional specimens.** 73 exx, Luozizuo Village (100.2173° E, 25.1142° N), Weishan County, Dali City, Yunnan, China, elev. 2364 m, 07.VII.2021, Yuchen Zhao and Xinpu Wang (SANXU); 35 exx, Qianchang Village (101.3704° E, 25.5372° N), Wuding County, Chuxiong City, Yunnan, China, elev. 2518 m, 23.V.2022, Yuchen Zhao (SANXU); 2♂♂, 3♀♀, Qianchang Village (101.3857° E, 25.5386° N), Wuding County, Chuxiong City, Yunnan, China, elev. 2181 m, 24.V.2022, Yuchen Zhao (SANXU); 18 exx, Madeng Village (99.9096° E, 26.5895° N), Jianchuan County, Dali City, Yunnan, China, elev. 2395 m, 01.VI.2022, Yuchen Zhao (SANXU); ♂, Jinniu Village (100.5142° E, 25.8351° N), Binchuan County, Dali City, Yunnan, China, elev. 1712 m, 02.VI.2022, Yuchen Zhao (SANXU); 13 exx, Jinniu Village (100.3089° E,25.7983° N), Binchuan County, Dali City, Yunnan, China, elev. 2102 m, 03.VI.2022, Yuchen Zhao (SANXU); ♂, Wande Village (102.1460° E,25.8312° N), Wuding County, Chuxiong City, Yunnan, China, elev. 1301 m, 11.VI.2022, Yuchen Zhao (SANXU); ♀, Wande Village (102.1931° E,25.9784° N), Wuding County, Chuxiong City, Yunnan, China, elev. 2133 m, 12.VI.2022, Yuchen Zhao (SANXU).


**Redescription.**


**Body.** Total length 1.72–2.20 mm.

**Color.** Slightly glossy, head and pronotum brownish-red; elytra yellowish-brown; mouthparts, antennae, legs yellowish (Figure 5A).

**Head.** Subtriangular, 0.95 times as long as wide, straighten posteriorly; frontoclypeal suture present; tempora as long as a diameter of eye; tempora slightly expansive toward head base, head base straight; posterior temporal angles of head distinct angled; eyes medium-sized, convex, interfacetal setae distinct; punctures of head large and dense, confluent, with some punctures squeezed into hexagonal shapes; setation decumbent, light, dense, point to median of head base; tactile setae indistinct (Figure 5B); antennae filiform, hardly extending to base of elytra; the first antennomere is slightly widened, antennomeres II–VI with similar length, VII–X gradually widen, X width wider than length; terminal antennomere elongate cylindrical, 2.5 times as long as the penultimate.

**Pronotum.** Pronotum 0.86 times as long as wide; pronotum obviously wider than head across eyes, with obvious anterior rim; lateral margins of pronotum angled in apical third, moderately narrowed posteriad, antebasal sulcus present but indistinct; punctures on pronotum as large as those on head, smaller on lateral side but still confluent, pubescence dense, yellow, appressed, pubescence directed posteriorly; tactile setae indistinct (Figure 5C).

**Scutellum.** Triangular, rounded distally.

**Elytra.** Glossy, 1.9 times as long as wide, postbasal impression absent, humeri distinct, broadly rounded; punctures slightly dense, with intervening spaces 1–2 times as large as punctures, punctures smaller than those on forebody; punctures gradually smaller and shallower toward apex; pubescence yellow, long, dense, decumbent, distinctly directed posteriorly; tactile setae indistinct (Figure 5J). Metathoracic wings fully developed.

**Legs.** Yellowish; femora slightly thickened distally but not stick-like; protibiae and metatibiae simple in both sex; basal metatarsomere hardly shorter than combined length of remaining metatarsomeres.

**Venter.** Glossy, yellowish; punctures of mesoventrite distinctly large and shallow; setation long, yellow, directed posteriorly; mesepisterna slightly separated by mesoventrite; mesepimera, posterior angle of mesoventrite and posterior margin of mesepisternum with fringed setation (Figure 5D); Last visible ventrite of male broad rounded with emarginated in middle distally (Figure 5E); distal margin of last visible tergite in male weakly emarginate (Figure 5F).

**Aedeagus.** Tegmen bilobated apically, lateral margin of tegmen slightly constricted toward apex then expands at the proximal tegmen, parameral plate 1.45 times as long as phallobase, lateral margin of penis apex uniformly contracted, somewhat anteriorly acute (Figure 5G,H).

**Sexual dimorphism.** Female externally similar to male while last visible ventrite broadly rounded distally.

**Variation.** Body length in the range of 1.98–2.24 mm. Some specimens collected from Yunnan, China, have darker heads and elytra, and their antennae gradually darken toward the apex.

**Ecology.** The habitat of *F. rubens* from China is similar to that of *F. maderi*. They were mostly collected by sweeping or beating grass on path beside forest or open space at elevations of 500–2600 m (Figure 9D,E).

**Distribution.** China (Yunnan), India (Arunachal Pradesh, Assam, Himachal Pradesh, Uttaranchal), Nepal, Pakistan.

**Figure 5 insects-14-00102-f005:**
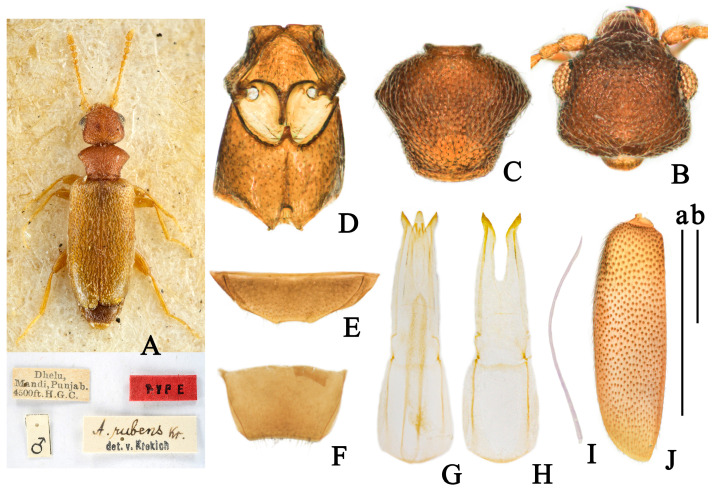
*Furcanthicus rubens* (Krekich-Strassoldo, 1931): (**A**) habitus and labels of holotype (photography by Dr. Harald Schillhammer); (**B**) head, male, dorsal view; (**C**) pronotum, male, dorsal view; (**D**) meso- and metasternum, male, ventral view; (**E**) ventrite VII, male, ventral view; (**F**) tergum VII, male, dorsal view; (**G**) aedeagus, dorsal view; (**H**) tegmen, dorsal view; (**I**) spiculum, male; (**J**) left elytron, male, dorsal view. Scale bar: (a) 0.5 mm for (**B**–**I**); (b) 0.5 mm for (**J**).

#### 3.3.2. *Furcanthicus validus* Zhao et Wang sp. nov.

(Figure 6)

urn:lsid:zoobank.org:act:327F9011-56C9-46BB-8AA7-AC8C2AD154C6

**Holotype.** ♂, Shimian County, Tibetan Autonomous Prefecture of Garzê, Sichuan, China, 09.VIII.2017, Zhangxun Wang (SANXU).

**Paratype.** ♀, same data as holotype (SANXU).


**Description.**


**Body. Holotype.** Total length 2.08 mm, maximum width 0.71 mm; head length 0.42 mm, head width across eyes 0.48 mm, pronotal length 0.42 mm, maximum pronotal width 0.51 mm, minimum pronotal width 0.33 mm, elytral length 1.22 mm, combined elytral width 0.71 mm.

**Color.** Brown, with forebody darker; mouthparts, antennae, and legs lighter, ventral side yellowish-brown (Figure 6A).

**Head.** Subsquare, 0.82 times as long as wide; frontoclypeal suture present; tempora as long as a diameter of eye, subparallel; head base subtruncate, with posterior temporal angles rounded; eyes medium-sized, convex, interfacetal setae short; punctures of head dense, nearly confluent, with some punctures squeezed into hexagonal shapes; setation decumbent, light and dense, point to median of head base; tactile setae indistinct (Figure 6B); antennae filiform, hardly extending to base of elytra; the first antennomere slightly widened, antennomeres II–VI with similar length, VII–X widened, penultimate antennomere width wider than length, terminal antennomere elongate cylindrical, 2 times as long as the penultimate.

**Pronotum.** Pronotum 0.82 times as long as wide; pronotum obviously wider than head across eyes, with anterior rim covered by head base, broadly rounded anteriorly, obvious lateral expansion in the apical third, moderately narrowed posteriad, antebasal sulcus present but indistinct; punctures of pronotum as large as those on head, smaller on lateral sides but still confluent; pubescence densely, yellow, decumbent, directed posteriorly. Tactile setae indistinct (Figure 6C).

**Scutellum.** Triangular, rounded distally.

**Elytra.** Glossy, 1.7 times as long as wide; postbasal impression absent; humeri distinct, broadly rounded; punctures of elytra dense, with intervening spaces shorter than one diameter of punctures, smaller than those on forebody, gradually smaller and shallower toward the apex; pubescence pale, dense, and decumbent, distinctly directed posteriorly; tactile setae indistinct (Figure 6J). Metathoracic wings fully developed.

**Legs.** Brown; femora slightly thickened distally but not stick-like; protibiae and metatibiae of males simple; basal metatarsomere hardly longer than combined length of remaining metatarsomeres.

**Venter.** Glossy, yellowish-brown; punctures of mesoventrite very large and irregular, the border of puncture confluent to ridge-shaped; setation of mesoventrite short, pale, directed posteriorly; mesepisterna slightly separated by mesoventrite; mesepimera, posterior angle of mesoventrite, and posterior margin of mesepisternum with fringed setation (Figure 6D).

**Aedeagus.** Tegmen bilobated apically, rolled into a cylindrical shape and open ventrally, parameral plate 1.8 times as long as phallobase, lateral margin of penis apex drastically contracted and anteriorly pointed (Figure 6G,H).

**Sexual dimorphism.** Last visible ventrite of male broad rounded with emarginated in middle distally (Figure 6E); distal margin of last visible tergite in male weakly emarginate (Figure 6F). Only one female specimen was examined in this study, it shows externally similar appearance to male while the last visible ventrite is broadly rounded distally.

**Diagnosis.** This species belongs to the *F. rubens* species-group, it is similar to *F. rubens* at morphological of male ventrite VII and tergite VII. It differs by its shorter pronotum (at least 0.86 times length as width in *F. rubens*), shorter elytra (at least 1.9 times length as width in *F. rubens*), and the morphology of male aedeagus.

**Etymology.** Named from the Latin “*validus*” [strong, mighty and valid] because of the wide body shape.

**Ecology.** Unknown.

**Distribution.** Only known from Tibetan Autonomous Prefecture of Garzê, Sichuan, China.

**Figure 6 insects-14-00102-f006:**
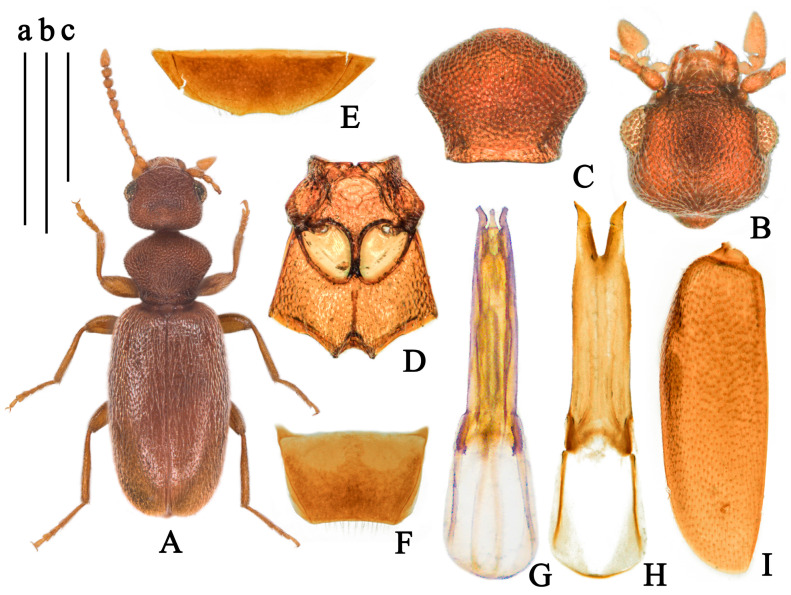
*Furcanthicus validus* sp. nov.: (**A**) habitus, female, dorsal view; (**B**) head, male, dorsal view; (**C**) pronotum, male, dorsal view; (**D**) meso- and metasternum, male, ventral view; (**E**) ventrite VII, male, ventral view; (**F**) tergum VII, male, dorsal view; (**G**) aedeagus, dorsal view; (**H**) tegmen, dorsal view; (**I**) left elytron, male, dorsal view. Scale bar: (a) 0.5 mm for (**A**); (b) 0.5 mm for (**B**–**H**); (c) 1 mm for (**I**).

### 3.4. Key to Species of Furcanthicus

1.Pronotum with width shorter than its length (Figure 1C), not wider than head across eyes (Figure 1A) ...... *F. maderi* species-group ................................................................... 2-Pronotum with width longer than its length (Figure 5C), wider than head across eyes (Figure 5A) ...... *F. rubens* species-group ............................................................................ 92.Elytra without obvious macule or band (Figure 2J) ........................................................ 3-Elytra with obvious orange or yellow transverse band near base (Figure 1J) ............. 63.Punctures on pronotum uniform, large, and dense and nearly confluent (Figure 2C); lateral margins of penis obvious narrowed distally, apex of penis pointed (Figure 2G) ................................................................................. *F. punctiger* (Krekich-Strassoldo, 1931)-Punctures on pronotum larger on disc, no confluent (Figure 3C); lateral margins of penis gradual narrowed distally, apex of penis rounded (Figure 3G) ......................... 44.Tactile setae of pronotum inconspicuous (Figure 3C); weakly emarginate distal margin of last visible tergite in male (Figure 3F) .................................. *F. acutibialis* **sp. nov.**-Tactile setae of pronotum conspicuously in lateral margins; strongly emarginate distal margin of last visible tergite in male (as Figure 4F) ................................................... 55.Tempora as long as a diameter of eye, almost not converging toward base; metathoracic wings reduced, significantly shorter than half of elytra length ...... *F. monstrator* (Telnov, 2005)-Tempora 1.7 times as long as a diameter of eye, converging toward base; metathoracic wings fully developed ....................................................... *F. vicarius* (Telnov, 2005)6.Eyes distinctly shorter than tempora (Figure 1B); elytra with shallow postbasal transverse impression ................................................................................................................... 7-Eyes slightly shorter than tempora (Figure 4B); elytra without postbasal transverse impression ............................................................................................................................. 87.Truncate distal margin of last visible tergite in male (see Figure 50 in Telnov, 2018 [29]); black area on base of elytra interrupted on suture ........... *F. vicinor* (Telnov, 2018)-Strongly emarginate distal margin of last visible tergite in male (Figure 1F); black area on base of elytra not interrupted on suture (Figure 1J) ............ *F. maderi* (Heberdey, 1938)8.Elytra suture yellow; lateral margins of penis rapidly narrowed distally, apex of penis pointed ....................................................................................... *F. lepcha* (Telnov, 2018)-Elytra suture with apical half black (Figure 4J); lateral margins of penis gradually narrowed distally, apex of penis rounded (Figure 4G) ........................ *F. telnovi* sp. nov.9.Pronotum with lateral margins angled in apical third (Figure 5C); eyes slightly shorter than tempora, tempora somewhat diverging towards head base, temporal angles squared (Figure 5B) ...................................... *F. rubens* (Krekich-Strassoldo, 1931)-Pronotum with lateral margins round apically (Figure 6C); eyes not shorter than tempora, tempora converging towards base, temporal angles rounded (Figure 6B) ...... *F. validus* sp. nov.

### 3.5. Species List and Distribution of Furcanthicus

(Figure 7)

*Furcanthicus acutibialis* Zhao et Wang sp. nov.

Distribution. China (Tibet).

*Furcanthicus lepcha* (Telnov, 2018) comb. nov.

Original placement. *Anthicus lepcha* Telnov, 2018:230

Distribution. India (Sikkim).

*Furcanthicus maderi* (Heberdey, 1938) comb. nov.

Original placement. *Anthicus maderi* Heberdey, 1938:163

Distribution. China (Yunnan), Thailand (Mae Hong Son).

*Furcanthicus monstrator* (Telnov, 2005) comb. nov.

Original placement. *Anthicus monstrator* Telnov, 2005:13

Distribution. China (Sichuan, Yunnan).

*Furcanthicus punctiger* (Krekich-Strassoldo, 1931) comb. nov.

Original placement. *Anthicus punctiger* Krekich-Strassoldo, 1931:23

Distribution. China (Yunnan), India (Sikkim, Uttaranchal), Nepal.

*Furcanthicus rubens* (Krekich-Strassoldo, 1931) comb. nov.

Original placement. *Anthicus rubens* Krekich-Strassoldo, 1931:20

Distribution. China (Yunnan), India (Arunachal, Assam, Himachal, Uttaranchal), Nepal, Pakistan.

*Furcanthicus telnovi* Zhao et Wang sp. nov.

Distribution. China (Yunnan).

*Furcanthicus validus* Zhao et Wang sp. nov.

Distribution. China (Sichuan).

*Furcanthicus vicarius* (Telnov, 2005) comb. nov.

Original placement. *Anthicus vicarius* Telnov, 2005:17

Distribution. China (Sichuan, Yunnan).

*Furcanthicus vicinor* (Telnov, 2018) comb. nov.

Original placement. *Anthicus vicinor* Telnov, 2018:232

Distribution. Nepal.

**Figure 7 insects-14-00102-f007:**
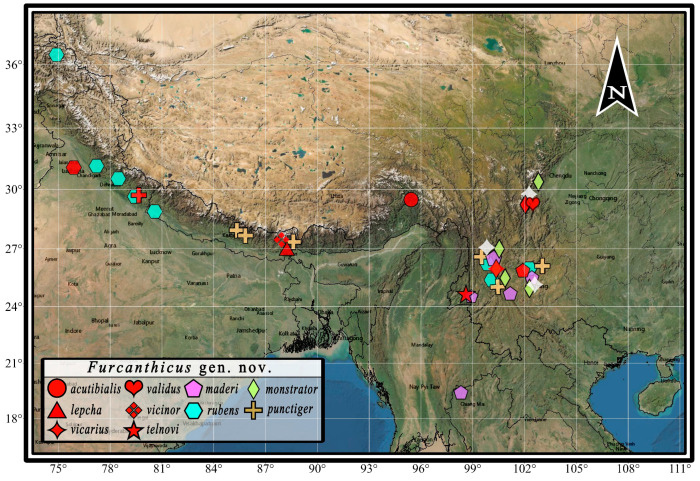
Geographic distribution of *Furcanthicus* gen. nov. species. Type locality of all species are marked in red.

Note:

*Nitorus lii* (Uhmann, 1997) comb. nov.

(Figure 8)

Original placement. *Pseudoleptaleus lii* Uhmann, 1997: 21

*Anthicus lii* (Uhmann, 1997) by Telnov, 2007: 81

**Type material.** Holotype ♂, China, Prov. Hebei Yongnian 6.11.1995 from cotton leg. Shuqiang Li [printed]//Pseudoleptaleus lii sp. n. det. G. Uhmann 1996 [printed] //Anthicidae, ohne No. ca. 20 individual 6-11.1995 Yongnian, China [printed]//Typus [printed, label red] (IZCAS).

**Additional specimen.** 1♂, Qingdao City, Shandong, China, 06.VII.2021, Rixin Jiang (SANXU).

**Distribution.** China (Hebei, Shandong).

**Remarks.** This species, collected from central China, was recorded as similar to *Anthicus maderi* based on the original description and drawings [21] and is here transferred to *Nitorus* Telnov, 2007 according to the type material.

This species with the characters combination of *Nitorus* [21]: Glossy and rarely pubescent body; pronotum with lateral margins distinctly constricted in basal half; anterior part of pronotum more convex than posterior; mesoventrite lateral margins straight; mesepisterna glossy, without fringe of setae, sparse covered with very fine whitish appressed pubescence. Moreover, this species has more characters typical of *Nitorus*, such as setae on elytron sparse, suberect; metasternum short and broad; first visible abdominal sternite simple, short, and broad, laterally nearly as long as the metasternum. While this species has a similar appearance to some *Furcanthicus* spp., they can be easily distinguished by the forms of the aedeagus (Figure 9B,C) and spiculum (Figure 9D).

**Figure 8 insects-14-00102-f008:**
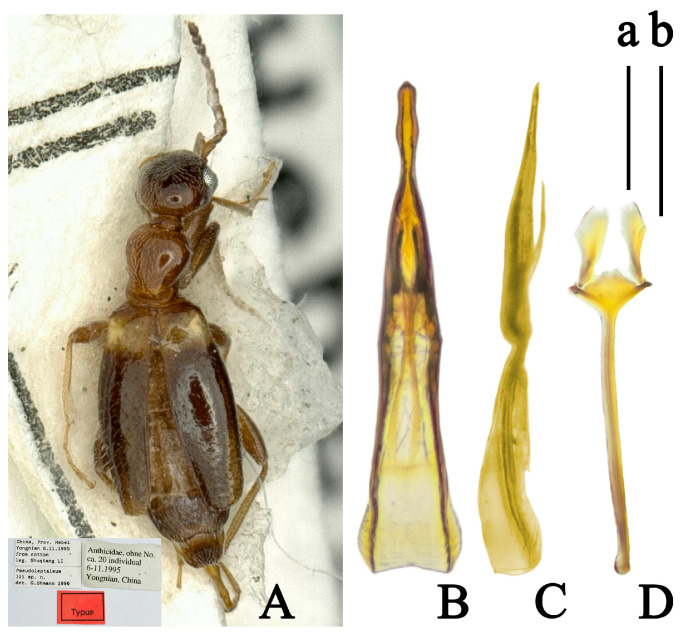
*Nitorus lii* (Uhmann, 1997): (**A**) habitus of holotype (photography by Dr. Kuiyan Zhang); (**B**) aedeagus, dorsal view; (**C**) aedeagus, lateral view; (**D**) spiculum, male. Scale bar: (a) 1 mm for (**A**); (b) 0.5 mm for (**B**–**D**).

**Figure 9 insects-14-00102-f009:**
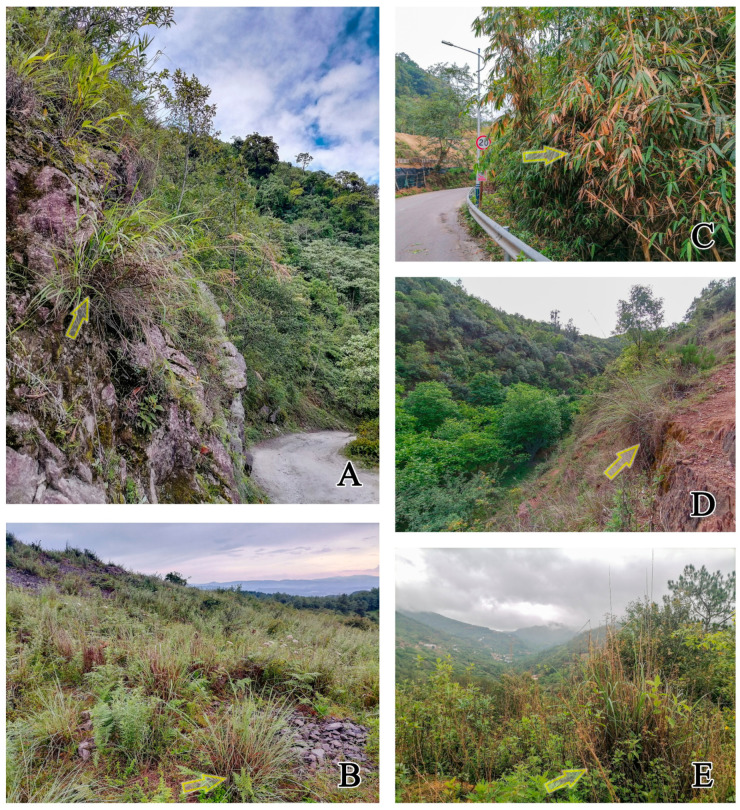
Habitat photos of *Furcanthicus* spp.: (**A**) Tibet, Mêdog county: *F. acutibialis*
**sp. nov.**; (**B**) Yunnan, Kunming, Xishan mountain: *F. monstrator* (Telnov, 2005) and *F. vicarius* (Telnov, 2005); (**C**) Yunnan, Baoshan, Longxing Village: *F. telnovi*
**sp. nov.** and *F. maderi* (Heberdey, 1938); (**D**) Yunnan, Chuxiong, Wuding county: *F. punctiger* (Krekich-Strassoldo, 1931) and *F. rubens* (Krekich-Strassoldo, 1931); (**E**) Yunnan, Dali, Weishan county: *F. maderi* (Heberdey, 1938), *F. punctiger* (Krekich-Strassoldo, 1931), and *F. rubens* (Krekich-Strassoldo, 1931). Collection sites indicated by arrows.

## 4. Discussion

This new genus primarily resembles *Sapintus* Casey, 1895, according to the following: (1) Pronotum with basal transverse sulcus continues laterally to foveae above procoxae; (2) mesepimera with conspicuous foveae, with setose fringe on margin of cavity; (3) abdominal ventrite III with intercoxal process pointed apically [3,26] but can be distinguish from: (i) *Furcanthicus* without foveae immediately behind metacoxae on ventrite III; (ii) *Sapintus* with setation of the elytra mostly heterogeneous, includes short, obliquely pointing undersetae; and (iii) *Furcanthicus* with phallobase basal margin rounded and tegmen bilobate apically. This genus also closely resembles *Ischyropalpus* LaFerté-Sénectère, 1849, from the New World due to the structure of mesoventrite, the phallobase being somewhat convex and basal margin rounded [33,34,35], *Furcanthicus* gen. nov. differs from it clearly in incompletely fused parameres (distinct separate parameres in *Ischyropalpus*) and parameres without setae on the apex (obvious setae in *Ischyropalpus*). In terms of other related genera in Anthicini, the genus differs from *Anthicus* in mesepimera due to more conspicuous foveae, extremely less sclerotized spiculum, and rounded phallobase basal margin. The difference between *Furcanthicus* gen. nov. and *Odacanthicus* Sakai & Ohbayashi, 1994, a subgenus of *Anthicus*, is that the new genus with process of mesoventrite is connected with metaventrite, and mesepisterna with foveae conspicuous (process of mesoventrite not reach the metaventral transverse carina, and mesepisterna reduced in *Odacanthicus*, furthermore, the male aedeagus similar to those of *Anthicus* [36]). *Furcanthicus* gen. nov. differs from *Acanthinus* LaFerté-Sénectère, 1849 in that mesepisterna slightly separated by mesoventrite and mesoventrite without fringe of evenly arranged setae on anterolateral margins (*Acanthinus* with mesepisterna touching medially [18] and mesoventrite bear fringe evenly arranged setae on anterolateral margins [26]); differs from *Cyclodinus* in that the mesepisterna with setose fringe on margin of cavity, without longer stiff setae near lateral angulation on mesepisterna, the aedeagus subcylindrical, the setae on forebody and elytra are longer and denser (*Cyclodinus* mesepisterna without setose fringe on margin of cavity, with anteriorly pointing longer stiff setae, aedeagus calamiforme [37], with setae on forebody and elytra short and sparse [25]); differs from *Cordicomus* by the setation of the mesepimera and the form of aedeagus (mesepimera with tuft of longer stiff setae on ventral side near lateral angulation, pointing anteriorly [17]; aedeagus calamiforme with tegmen apex hooks, teeth, or lateral expansions in *Cordicomus* [37]); *Furcanthicus* gen. nov. similar to *Nitorus* Telnov, 2007, and some *Stricticomus* Pic, 1894, with the body being glossy and pronotum having lateral margins distinctly constricted in the basal half, but they can be distinguished from *Furcanthicus* by the pronotum with a conspicuous lateral pit, thin and extremely less sclerotized spiculum and rounded phallobase basal margin.

## Data Availability

All data presented in this paper are included within the article and are available for use. All nomenclatural acts has been registered with ZooBank (https://zoobank.org/ accessed on 1 November 2022).

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
