# Peer review of "Furcanthicus gen. nov., a New Genus of Oriental Anthicini (Coleoptera, Anthicidae), with Description of Three New Species†"

_insects, 2023, doi:10.3390/insects14020102_

Round 1

Reviewer 1 Report (Previous Reviewer 1)

The present manuscript on a new genus and several new species of the Anthicini improved significantly since the first review.

However, there are still severe inconsistency appearing through the text as well as the diagnosis of a new genus remains insufficient from the scientific point of view.

The diagnosis is improved but, unfortunately, not sufficient to justify erection of a new genus. The authors BELIEVE this genus is new, but this is not scientific and not professional to claim that 'these characters ... but never in this combination '

The authors have to provide scientific evidence of the differences between Furcanthicus and similar genera (for instance but not only, Acanthinus, Cordicomus, Cyclodinus, Ischyropalpus, Sapintus, Stricticomus etc.) highlighting the differences of evolutionary value to justify the new genus and avoid using features of dubious phylogenetic value like shape of pronotum or head as was done by earlier authors.

There are several inconsistencies found in the (re)descriptions and the key. Some of them, but not all, highlighted in the reviewed document. 

Some unclear or wrong statements found in morphological descriptions are 'inner side of tibia' (which reads inside tibia; it is not fully clear to me what is meant here, possibly posterior margin of tibia), '1 times as large' (which should be rephrased since it means as large as) etc. (see text for items marked 'unclear' by the reviewer).

English improved but still requires moderate changes since several statements and sentences appear unclear, misleading or grammatically incorrect. I highlighted some but not all of them. Some of common mistyping in the text are use of "obvious' instead of distinct, 'toward' instead of towards, 'puncture' instead of punctures, 'bases' (of tibiae) instead of base (only one base present in each tibia!), 'with puncture' instead of punctures etc. The manuscript to be carefully read and corrected.

There are several other comments in the reviewed text to be considered and addressed.

The manuscript indeed remains interesting and is recommended for the publication after another major revision.

Author Response

Dear reviewer:

Thank you for your sincere opinions on this manuscript! We modify the manuscript according to your review suggestions. Please download the attachment to view our response.

Thank you for reviewing this manuscript, and wishing you all the best.

Sincerely,
Authors
05 January 2023

Reviewer 2 Report (New Reviewer)

manuscript should be corrected mostly in some phrases, current morphological terminology, proper formation in Italic or Bold. Most corrections were made in the file attached

Author Response

Dear reviewer:

Thank you for your sincere opinions on this manuscript! We modify the manuscript according to your review suggestions. Please download the attachment to view our response.

Thank you for reviewing this manuscript, and wishing you all the best.

Sincerely,
Authors
05 January 2023

Round 2

Reviewer 1 Report (Previous Reviewer 1)

The manuscript has been improved significantly since the last review. The diagnosis of the new genus has to be supplemented and comparision of it with Odacanthicus should be provided. Authors intentionally or intentionally overlooked this taxon in the diagnosis. Thoracic structures are not described in literature for Odacanthicus and remain unknown. This taxon appears externally similar to Furcanthicus. It is therefore necessary to supplement the differential diagnosis. After the diagnosis is supplemented, the manuscript can be accepted for publication. 

Author Response

Dear reviewer:

Odacanthicus Sakai & Ohbayashi, 1994 is a monotypic subgenus from Japan, and it does indeed resemble the new genus in appearance. We have compared Furcanthicus and Odacanthicus in the Discussion section according to your suggestion.

Thank you for reviewing this manuscript, and wishing you all the best.

Sincerely,
Authors
14 January 2023

This manuscript is a resubmission of an earlier submission. The following is a list of the peer review reports and author responses from that submission.

Round 1

Reviewer 1 Report

The present manuscript is focused on several new descriptions and redescriptions of Palaearctic Anthicini, including a new genus.

The English language of the manuscript is often confusing or fully incorrect (from scientific point of view, in particular). Many statements require rephrasing to become clear. The manuscript to be proof-read by a native English speaker. See some comments on this issue in the text.

I therefore only highlighted some linguistic problems, not all of them and did not check all species (re)descriptions, but only the first one. The same or similar issues are, indeed, distributed all throughout the text and not only those which are highlighted by me requires corrections.

The authors appear to be new in beetle taxonomy (or this is the problem of their English), since numerous statements are wrong from the scientific point of view (for instance, their confused “apical” with “basal”, as on row 231). This requires proper check by a specialist in beetle morphology. See several comments on this issue in the text.

In the Material and Methods the authors stated that they follow terminology from works of Chandler and Kejval & Chandler. But this is not true – the authors use some odd / not commonly used terms in their descriptions. The authors should follow the mentioned works or either delete the statement that they use terminology from aforementioned specialists.

The authors placed their Key before actual descriptions of the new species and before formal introduction of the new names. This is very bad tone since these names, when they appear without description / type designation in the paper, could be considered nomina nuda. I recommend moving the Key to the Discussion.

The morphological features used in the Key are mainly colour-based. The colour pattern varies greatly in Anthicini and relying on the colour should be avoided. Moreover, in the Key some features are only listed for one part of the same coupled and not repeated in the second part of the same couplet. This must be corrected. The Key as is could not be properly used, see notes in the text.

The description of the new genus is incomplete. On one hand, some features are mentioned which are common in all Anthicidae (which has little taxonomic value in this case). On the other hand, the most important feature - the structure of pro- and mesothorax - is described poorly and inaccurate (this might be the language issue, of course) and requires additions and corrections. See notes in the text.

The authors are not known experts in anthicid beetles taxonomy, compare their new genus basically only with Ischyropalpus ignoring a wide range of over 25 other Anthicini genera. External similarity of some of the discussed species with Ischyropalpus is based on features of questionable evolutionary value (e.g., denticulate tibiae or shape of aedeagus). Similar features appear randomly in different Anthicini genera, not only in Ischyropalpus and Furcanthicus. The authors in the diagnosis of the new genus have to provide comprehensive and clear assessment of all similar genera of Anthicini and not only compare Furcanthicus with Ischyropalpus ignoring or omitting major genera like Anthicus, Cordicomus, Sapintus, Cyclodinus, Stricticomus, Nitorus etc. From this diagnosis is it fully unclear how the newly proposed genus is different from other Anthicini subgroups!

Similarly, in the Discussion, the authors only focus on Ischyropalpus and completely ignoring other, possibly phylogenetically much more related, subgroups.

The authors are wrongly using “ = “ symbol, which in taxonomy should be used for indicating synonyms or homonyms. But the authors used “ = “ for original placements of the species, which is incorrect and could destabilize the nomenclature (since it appears like proposals of new synonymy which it is not). See comments in the text.

The authors are not known experts in anthicid beetles taxonomy, therefore the proposed new combination for Anthicus lii (Uhmann) requires additional justification (see comment in the text). The proposed new combination itself is indeed correct.

One of the “new” species is of the authors is in fact already described decades ago from Indochina Peninsula and currently attributed to Sapintus. The authors are at high risk of establishing new synonymy.

Generally talking, this highly interesting manuscript requires major revision and I will be happy to review it for the second time.

Respectfully,

The Reviewer

Author Response

Dear reviewer,

Thank you for your review and suggestions. We have provide a point-by-point response to your comments. Please see the attachment for details.

Sincerely,
Yuchen Zhao

Reviewer 2 Report

Interesting paper! Authors nicely recognized a distinctive group within Anthicini, photos are fine, key surely useful (character of tegmen could be also included). The main problem is rather vague definition of new genus. It should be clearly stated what are those unique or at least important character. Similarly, I can not see clear evidence for new placement of Anthicus lii. This should be undoubtedly improved. Numerous small problems in English usage noted (some indicated in redescription of F. maderi) and also in terminology (indicated within ms). Otherwise I can recommend for publication and would be later grateful for sending pdf.

Author Response

(The authors gave the same response as above.)
